# Dissection of zebrafish *shha* function using site-specific targeting with a Cre-dependent genetic switch

Kotaro Sugimoto[1], Subhra P Hui[1], Delicia Z Sheng[1], Kazu Kikuchi[1,2]*

[1]Developmental and Stem Cell Biology Division, Victor Chang Cardiac Research Institute, Darlinghurst, Australia; [2]St. Vincent's Clinical School, University of New South Wales, Kensington, Australia

**Abstract** Despite the extensive use of zebrafish as a model organism in developmental biology and regeneration research, genetic techniques enabling conditional analysis of gene function are limited. In this study, we generated *Zwitch*, a Cre-dependent invertible gene-trap cassette, enabling the establishment of conditional alleles in zebrafish by generating intronic insertions via in vivo homologous recombination. To demonstrate the utility of *Zwitch*, we generated a conditional *sonic hedgehog a* (*shha*) allele. Homozygous *shha* mutants developed normally; however, *shha* mutant embryos globally expressing Cre exhibited strong reductions in endogenous *shha* and *shha* target gene mRNA levels and developmental defects associated with null *shha* mutations. Analyzing a conditional *shha* mutant generated using an epicardium-specific inducible Cre driver revealed unique roles for epicardium-derived Shha in myocardial proliferation during heart development and regeneration. *Zwitch* will extend the utility of zebrafish in organ development and regeneration research and might be applicable to other model organisms.

*For correspondence: k.kikuchi@ victorchang.edu.au

**Competing interests:** The authors declare that no competing interests exist.

## Introduction

Conditional gene knockout (cKO) is a genetic technique that enables the investigation of gene function in a temporally regulated manner in specific tissues or cell types (*Lobe and Nagy, 1998*; *Rajewsky et al., 1996*; *Rossant and McMahon, 1999*). Although the global KO approach is useful for investigating the first developmental function of a gene of interest, a conditional approach is often essential for deciphering the tissue-specific function of a gene in later biological events, avoiding the confounding effects of embryonic lethality or other morphogenetic defects that can result from global gene deletion. The cKO approach has been used most extensively in mice, owing to the availability of embryonic stem (ES) cells. ES cells possess the unique capacity of efficient homologous recombination (HR) (*Koller and Smithies, 1992*; *Thomas and Capecchi, 1987*), thereby providing a platform for engineering loxP-flanking (floxed) genomic segments that can be excised by Cre recombinase. However, recent advances in genome editing technologies have provided new possibilities for efficiently engineering conditional alleles in any model organism without the need for ES cells (*Bedell et al., 2012*; *Brown et al., 2013*; *Dickinson et al., 2013*; *Wang et al., 2013*).

The zebrafish has been widely used as a model organism in developmental biology and regeneration research. However, the first floxed allele of an endogenous zebrafish gene generated using transcription activator-like effector nuclease (TALEN)-mediated HR was only recently reported (*Hoshijima et al., 2016*). To expand the range of genetic tools that can facilitate cKO analysis in zebrafish, we explored an alternative approach using a Cre-dependent genetic switch referred to as *FLEx* (*Schnütgen et al., 2003*). *FLEx* is characterized by strategically arranged wild-type (WT) and mutant loxP sites that facilitate the induction of Cre-dependent stable inversion of a gene trap

cassette (*Schnütgen et al., 2003*). Although similar approaches have been used for conditional gene trap mutagenesis in zebrafish (*Clark et al., 2011*; *Ni et al., 2012*; *Jungke et al., 2016*; *Trinh et al., 2011*), no reports have described the precise targeting of a *FLEx* cassette at defined loci. In the present report, we describe a streamlined method to generate a conditional allele using a *FLEx*-based genetic switch via in vivo HR. Using this approach, we successfully generated a targeted *sonic hedgehog a* (*shha*) cKO allele to analyze *shha* function during heart development and regeneration. Our findings demonstrate the utility of the *FLEx*-based method for cKO analysis of gene function in zebrafish.

## Results

### Insertion of an invertible gene trap cassette via in vivo genome editing

We modified FT1 (*Ni et al., 2012*), a *FLEx* construct used for random gene trap mutagenesis in zebrafish, and generated a donor vector for the generation of zebrafish with an invertible gene trap cassette via HR (*Zwitch*) (*Figure 1A* and *Figure 1—figure supplement 1A*). *Zwitch* consists of a removable, lens-specific enhanced green fluorescence protein (EGFP) tag (LG) (*Lee et al., 2005*) and an invertible splice acceptor site conjugated to a gene expressing a red fluorescent protein (TagRFP) via a 2A self-cleaving peptide sequence (*Figure 1A* and *Figure 1—figure supplement 1B*). The region encompassing LG and the gene trap cassette is flanked by unique restriction enzyme sites in which the right arm (RA) and left arm (LA) of homologous sequences are inserted (*Figure 1A* and *Figure 1—figure supplement 1A*).

We selected *shha* (GenBank ID: 30269), the gene encoding Sonic Hedgehog a, to test the utility of *Zwitch*. We tested several pairs of TALENs and selected the one that most efficiently induced DNA double-strand breaks (DSBs) in intron 1 of *shha* (*Figure 1B and C*). mRNA encoding the TAL-ENs was co-injected with the donor vector (pZwitch-shha-int1) into one-cell-stage embryos. The donor vector includes the RA and LA sequences in the orientation that facilitated the insertion of *Zwitch* in the non-mutagenic orientation (*Figure 1D*). The injected fish were examined for LG expression at 7 and 45 days post-fertilization (dpf), and fish that maintained LG expression at 45 dpf were raised to adulthood and outcrossed with WT fish (*Figure 1E*). We screened 32 LG$^+$ potential founders, finding that 19 of them (59%) transmitted LG expression (*Figure 1E*). PCR demonstrated that *Zwitch* was inserted in the correct non-mutagenic orientation in 17 (89%) of the 19 LG$^+$ founders (*Figure 1E and F*). The precise location of the insertion was further verified using DNA sequencing (*Figure 1—figure supplement 2A and B*) and Southern blot analysis (*Figure 1G*). We subsequently established a conditional gene trap line, *Tg(shha:Zwitch)$^{vcc8Gt}$*, which is referred to as *shha$^{ct}$* hereafter. We explained key steps for using *Zwitch* for other genes in the subsection titled 'Targeting *Zwitch* into other genes' in the Materials and methods.

### Characterization of the conditional *shha* allele

To characterize the *shha$^{ct}$* allele (*Figure 2A*), we injected one-cell-stage *shha$^{ct/+}$* embryos with a Cre expression vector (pUbb-iCRE-GFP; *Figure 2—figure supplement 1A and B*). Genomic PCR and reverse transcription (RT)-PCR detected the gene trap cassette inversion (*Figure 2B*) and alternative *shha* splicing in embryos injected with Cre DNA (Cre$^+$) but not in uninjected embryos (Cre$^-$) (*Figure 2C*). DNA sequencing verified the precise inversion (*Figure 2—figure supplement 2A*) and in-frame transcription of *shha-P2A-TagRFP* mRNA (*Figure 2—figure supplement 2B*). We injected Flp mRNA into one-cell-stage embryos of the offspring of an incross of *shha$^{ct/+}$* fish and detected the excision of the FRT-flanked LG cassette using genomic PCR (*Figure 2D and E*). EGFP fluorescence was lost or reduced at approximately the expected frequency of the homozygous or heterozygous *shha$^{ct}$* genotype (*Figure 2F and G*).

To validate that *shha$^{ct}$* was a loss-of-function allele, we obtained *shha$^{ct/ct}$* embryos from incrosses of *shha$^{ct/+}$* fish and individually genotyped the embryos using PCR (*Figure 3—figure supplement 1A and B*). Quantitative RT-PCR (qRT-CPR) revealed that the expression of *shha* and Hedgehog (Hh) target genes was strongly decreased in Cre$^+$*shha$^{ct/ct}$* embryos but not in Cre$^-$ *shha$^{ct/ct}$* embryos, indicating that the *shha$^{ct}$* allele is functional in the absence of Cre activation (*Figure 3A*). Furthermore, most Cre$^+$ *shha$^{ct/ct}$* embryos exhibited truncated pectoral fins and U-shaped somites that lacked clear horizontal myosepta (*Figure 3B and C*), defects reminiscent of homozygous *shha*

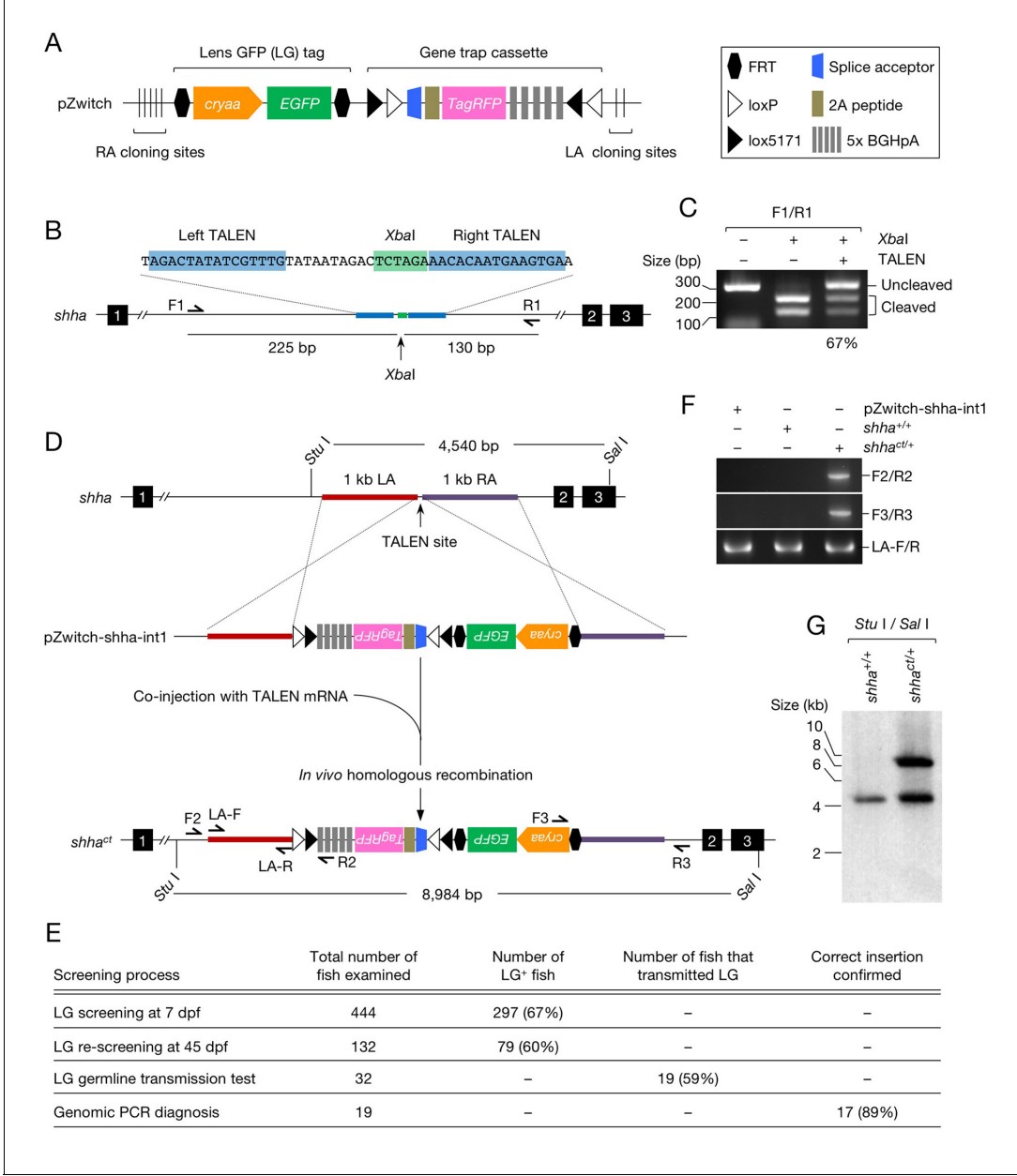

**Figure 1.** Generation of the *shha* conditional allele using *Zwitch*. (**A**) Schematic of *Zwitch*. (**B**) Schematic of the zebrafish *shha* locus and TALEN used to induce DNA DSBs in intron 1. Exons are indicated by filled boxes with numbers. The binding sites for the TALEN pair are highlighted in blue, and the *Xba*I site in the spacer region is highlighted in green. (**C**) The efficiency of the TALENs in introducing DSBs. *Xba*I digestion of PCR products amplified from the genomic DNA of embryos injected with TALEN mRNAs. The efficiency of the TALEN pair in inducing DSBs (67%) was quantified from the gel image using ImageJ software. (**D**) Schematic of the strategy used to target *shha* via TALEN-mediated homologous recombination with pZwitch-shha-int1. (**E**) The screening process for founders. (**F**) Genomic PCR analysis of the *Zwitch* insertion with the correct orientation. (**G**) Southern blot analysis of the *Zwitch*-modified *shha* allele. BGHpA, bovine growth hormone polyadenylation signal; *cryaa*, α A-crystallin; LA, left arm; RA, right arm.
The following figure supplements are available for figure 1:

**Figure supplement 1.** pZwitch vector.
**Figure supplement 2.** DNA sequence of the *shha^{ct}* allele.

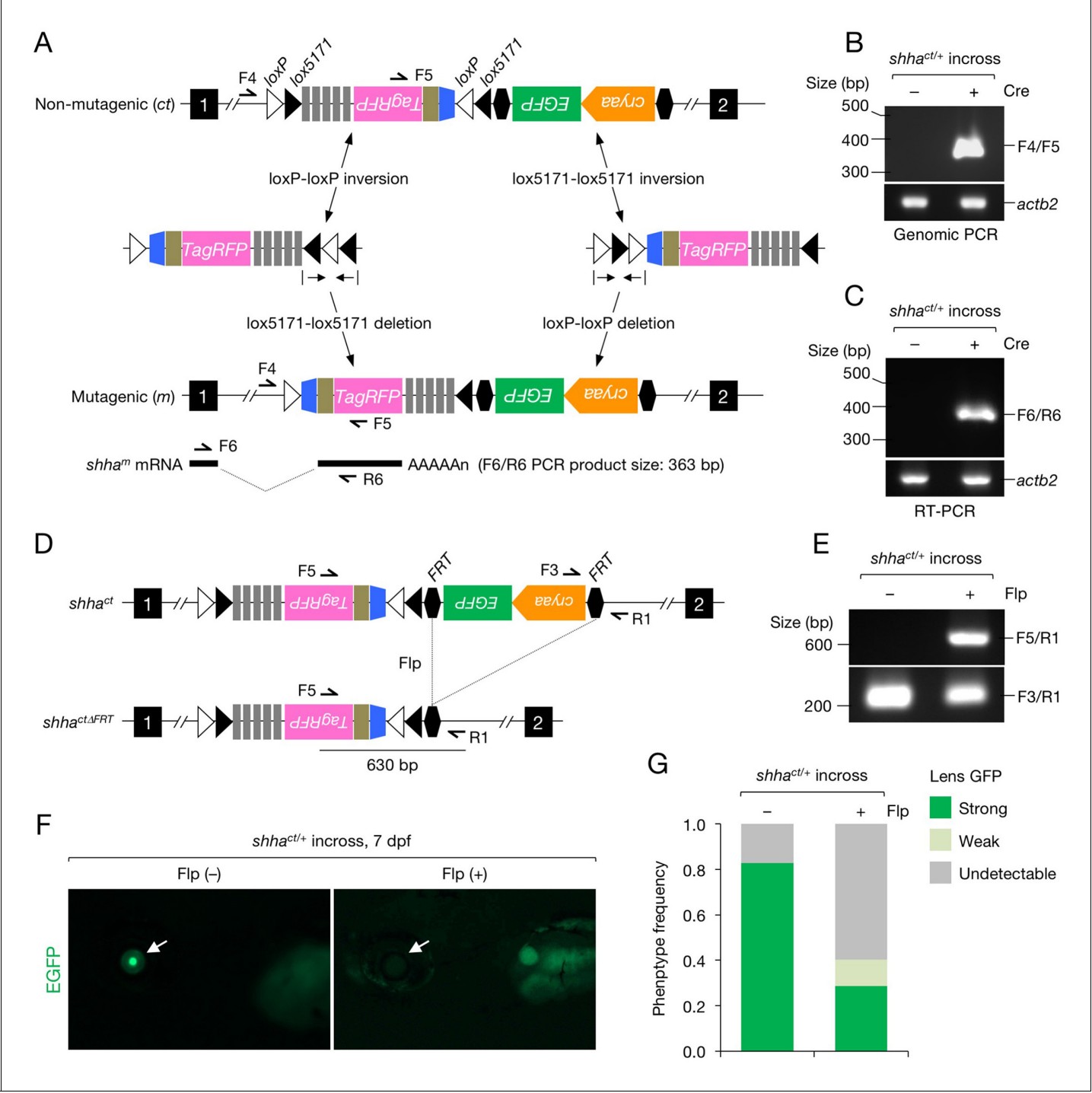

**Figure 2.** Characterization of the *shha^ct* allele. (**A**) Schematic of Cre-dependent conversion of *Zwitch* from the non-mutagenic orientation to the mutagenic orientation. Cre activation induces an inversion between loxP or lox5171 sites and the subsequent excision of loxP or lox5171-flanking DNA sequences (***Schnütgen and Ghyselinck, 2007***), thereby permanently converting *Zwitch* into the mutagenic form and inducing aberrant *shha* splicing. (**B**) PCR analysis of the *Zwitch* inversion. Genomic DNA from 72 hpf Cre+ and Cre− *shha^ct/+* embryos was analyzed using PCR. (**C**) RT-PCR analysis of *shha* expression in 72 hpf Cre+ and Cre− *shha^ct/+* embryos. (**D**) Schematic of Flp-mediated excision of the FRT-flanked LG tag in the *shha^ct* allele. (**E**) Genomic PCR analysis of Flp-injected (+) or uninjected (−) embryos from a cross of *shha^ct/+* adults. PCR using F3 and R1 primers detected *shha^ct* alleles in both samples. Flp mRNA was synthesized from linearized pCS2-FLPo (Materials and methods). (**F**) Representative image of embryos injected with Flp mRNA. Arrows indicate the lens. (**G**) Quantification of phenotypes of the embryos analyzed in **F**. A total of 87 Flp-injected (+) and 99 uninjected embryos (−) were analyzed (****$p < 1.0 \times 10^{-8}$ Fisher's exact test). dpf, days post-fertilization.

*Figure 2 continued on next page*

*Figure 2 continued*

The following figure supplements are available for figure 2:

**Figure supplement 1.** Cre expression vector.

**Figure supplement 2.** DNA sequence of the inverted *shha^ct^* allele and its transcript.

**Figure supplement 3.** TagRFP expression from the inverted *shha^ct^* allele.

mutant embryos (*Schauerte et al., 1998*). To confirm this result, we established *shha^ct/+^* fish carrying the transgene *Tg(ubb:iCRE-GFP)* (*ubb:Cre-GFP*), in which codon-improved Cre (iCRE) DNA was expressed by a strong, ubiquitously expressed *ubiquitin B* (*ubb*) promoter (*Mosimann et al., 2011*) (*Figure 2—figure supplement 1A*). From incrosses of this line, we obtained embryos carrying the WT and/or mutagenic *shha* allele and analyzed the phenotype of these embryos (*Figure 3—figure supplement 2A and B*). We found that pectoral fin development was largely normal in WT and heterozygous mutant embryos but severely hampered in all homozygous mutants examined (*Figure 3—figure supplement 2C*). We also performed semi-qRT-PCR analysis of *shha* expression and confirmed that its expression was reduced to nearly 50% of WT levels in the heterozygous mutants and

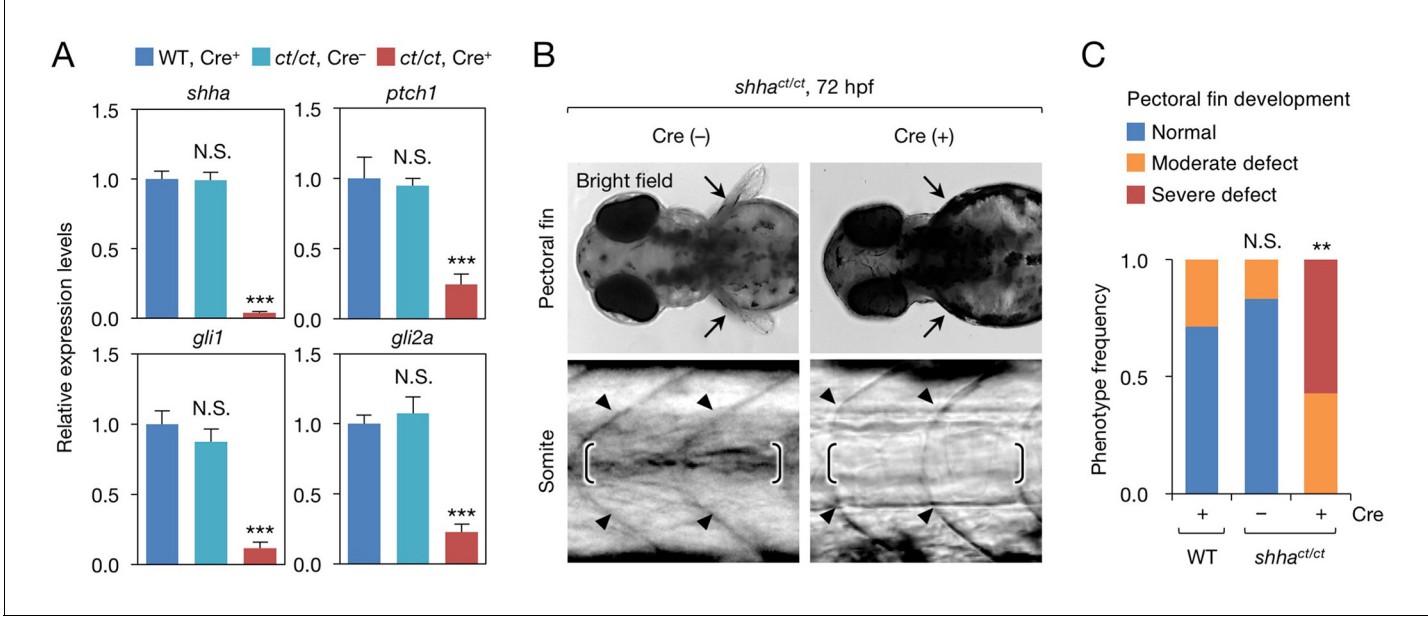

**Figure 3.** Phenotype of *shha^ct/ct^* embryos globally expressing Cre. (A) qRT-PCR analysis of 72 hpf Cre^+ and Cre^− *shha^ct/ct^* embryos (*n* = 10 and 9). WT embryos injected with pUbb-iCRE-GFP DNA were used as a control (*n* = 9). Ten pooled embryos per sample were used for qRT-PCR analysis. The data are presented as the mean ± SEM (***p<0.001, Mann–Whitney U test). (B) Phenotypes of 72 hpf Cre^+ and Cre^- *shha^ct/ct^* embryos. Arrows, pectoral fins; arrowheads, somite boundaries; brackets, horizontal myoseptum. Bright field images were captured using an MVX10 microscope. The composite images shown were generated using ImageJ software. Somite defects were observed in all embryos with severe pectoral fin defects. (C) Quantification of pectoral fin phenotypes from the embryos in B (*n* = 7 [WT, Cre^+], *n* = 12 [*shha^ct/ct^*, Cre^−], and *n* = 28 [*shha^ct/ct^*, Cre^+]; **p<0.01, Fisher's exact test). N. S., not significant (p=0.5392). The embryos used in A–C were selected on the basis of their high-level expression of Cre as described in *Figure 2—figure supplement 1B* (see also Cre DNA and mRNA injection, Materials and methods). A moderate pectoral fin defect was observed in control samples, likely due to injection artifacts.

The following figure supplements are available for figure 3:

**Figure supplement 1.** Genotyping PCR of *shha^ct^* alleles.

**Figure supplement 2.** Analysis of embryos carrying mutagenic *shha* alleles.

to an undetectable level in the homozygous mutants (*Figure 3—figure supplement 2D and E*). Together, these results provide clear evidence that *Zwitch* can inducibly disrupt gene function in zebrafish.

## Epicardium-specific inactivation of *shha* expression during zebrafish heart development

To demonstrate the utility of *Zwitch*, we investigated the functional consequences of inactivation of *shha* expression in the epicardium, the mesothelial layer covering the heart. Although the epicardium was reported to produce Shh during heart development in mice (*Lavine et al., 2006*), the precise function of Shh has not been addressed using an epicardium-specific KO approach. Using fluorescence-activated cell sorting (FACS), we purified epicardial cells and cardiomyocytes from the dissected hearts of transgenic reporter fish (see Flow cytometry, Materials and methods) and confirmed *shha* expression in epicardial cells and *ptch1* expression in both epicardial cells and cardiomyocytes (*Figure 4A*). In situ hybridization analysis also detected *shha* expression in the epicardium of the ventricle (*Figure 4—figure supplement 1A–C*). Next, we crossed *shha*$^{ct/+}$ fish with *shha*$^{ct/+}$ fish carrying the epicardium-specific inducible Cre transgene *Tg(tcf21:CreER)* (*tcf21:CreER*) (*Kikuchi et al., 2011*) and obtained *tcf21:CreER; shha*$^{ct/ct}$ embryos after PCR genotyping of individual embryos (*Figure 3—figure supplement 1C*). We inactivated *shha* expression in the epicardium by treating 24–72 hr post-fertilization (hpf) embryos with 4-hydroxytamoxifen (4-HT; *Figure 4B*), which induced a nearly complete inversion of the gene trap cassette (*Figure 4—figure supplement 2D*). Strikingly, most embryos lacking *shha* expression in the epicardium (epi-KO) developed severe cardiac edema by 96 hpf (*Figure 4C*). Cardiac edema was unclear or extremely weak in epi-KO hearts at 72 hpf, suggesting that epicardial inactivation of *shha* expression leads to heart defects at later developmental stages. We could not determine whether a similar cardiac phenotype was also observed in the global *shha* mutant embryos, as proper characterization was not possible due to the pleiotropic effects of global inactivation of *shha* expression at later time points. The epi-KO embryos had a thinner myocardial wall (*Figure 4D*) and significantly fewer cardiomyocytes (*Figure 4E*). These findings indicate that epicardium-derived Shha plays a crucial role in zebrafish heart development.

To investigate the mechanism underlying the developmental defects in epi-KO hearts, we analyzed the expression of the pan-epicardium markers *tcf21* and *pard3* using semi-qRT-PCR (*Plavicki et al., 2014*; *Serluca, 2008*) and detected the normal expression of these marker genes in epi-KO hearts (*Figure 4F*). This result suggests that epicardium-specific loss of *shha* does not directly affect the development of the epicardium, consistent with the observation that epicardial development is normal in mutant mouse hearts lacking the Shh receptor smoothened (*Rudat et al., 2013*). However, the expression of the well-characterized epicardial marker *raldh2/aldh1a2*, which encodes an enzyme that plays a key role in retinoic acid (RA) synthesis (*Begemann et al., 2001*; *Cunningham and Duester, 2015*; *Niederreither et al., 1999*; *Sucov and Evans, 1995*), was strongly decreased (*Figure 4F*). To analyze epicardial cell development and *raldh2* expression in epi-KO hearts, we crossed *tcf21:CreER; shha*$^{ct/+}$ with *shha*$^{ct/+}$ fish carrying the epicardium-specific DsRed2 reporter transgene *Tg(tcf21;DsRed2)* (*tcf21:DsRed2*) (*Kikuchi et al., 2011*) and obtained *tcf21: DsRed2; tcf21:CreER; shha*$^{ct/ct}$ embryos after PCR genotyping of individual embryos. We induced epicardium-specific inactivation of *shha* expression in these embryos and found that DsRed2$^+$ epicardial cells were normally distributed throughout the epi-KO heart (*Figure 4G*). Consistent with the PCR results (*Figure 4F*), Raldh2 protein expression was barely detectable in mutant DsRed2$^+$ cells (*Figure 4G*). These findings suggest that defects in RA synthesis might underlie the developmental defect in epi-KO hearts.

The expression of *fgf2* and *wnt9b*, two cardiomyocyte mitogens regulated by RA signaling during mouse heart development (*Merki et al., 2005*), was strongly decreased in epi-KO hearts (*Figure 4H*), whereas that of *igf2a* and *igf2b*, factors regulated by liver-derived erythropoietin (*Brade et al., 2011*), was unaffected (*Figure 4H*). Interestingly, the expression of *fgf9*, a gene encoding a key FGF regulated by RA signaling in the mouse heart (*Lavine et al., 2005*), as well as *fgf20a* and *fgf20b* was unaffected in epi-KO hearts, whereas *fgf16* expression levels was strongly decreased (*Figure 4H*). These findings suggest that although FGF regulation by RA signals is evolutionary conserved, different FGF proteins may regulate cardiogenesis in zebrafish and mammals. We also analyzed the expression of *fgf2*, *fgf16*, and *wnt9b* in purified cardiomyocytes and epicardial cells (*Figure 4A*), finding that they are primarily expressed in the epicardium during zebrafish heart

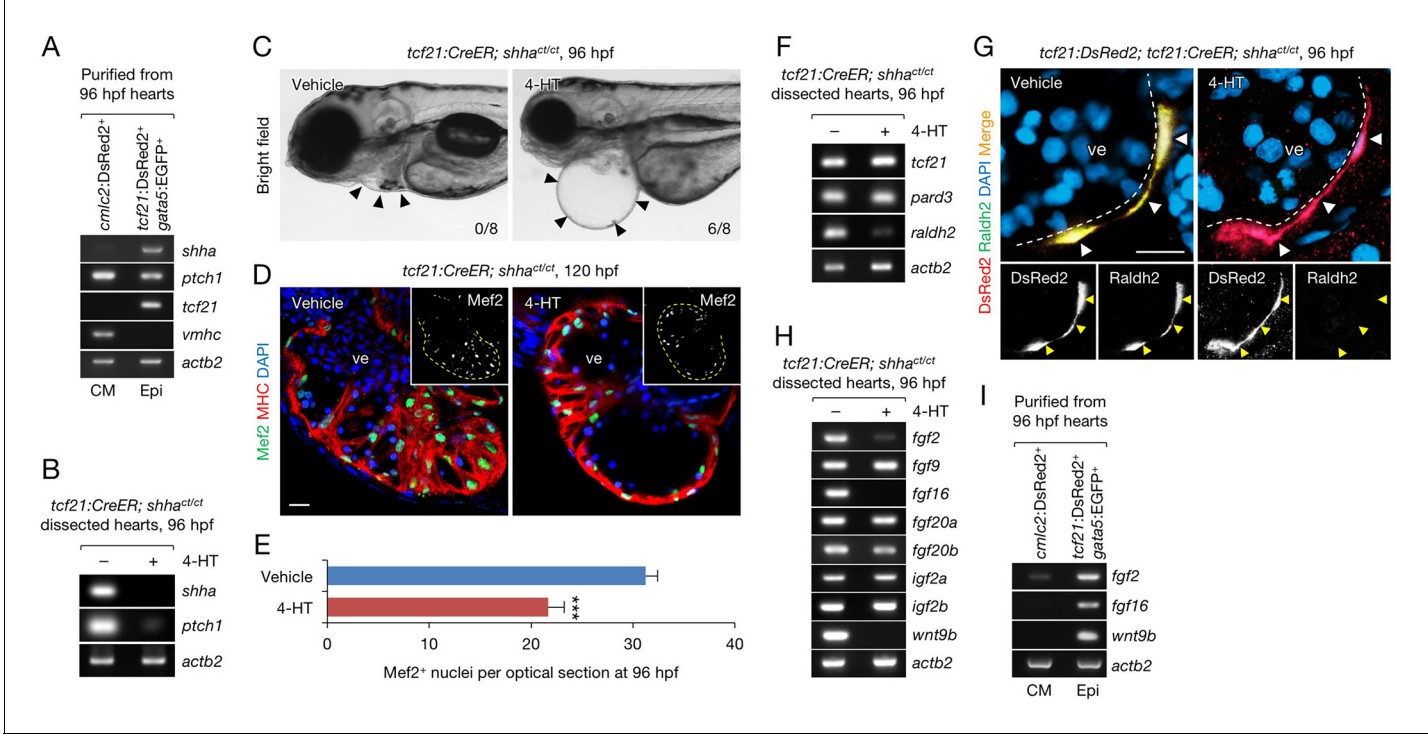

**Figure 4.** Epicardium-specific inactivation of *shha* expression during heart development. (**A**) Semi-qRT-PCR analysis of purified cardiomyocytes (CM) and epicardial cells (Epi) from 96 hpf hearts (see also Flow Cytometry, Materials and methods). Cardiomyocyte (*vmhc*) and epicardial (*tcf21*) markers were used to confirm the specificity of cell sorting. (**B**) Semi-qRT-PCR analysis of *shha* and *ptch1* expression in hearts dissected from *tcf21:CreER; shha^{ct/ct}* embryos treated with the vehicle (−) or 4-HT (+). (**C**) Phenotype of *tcf21:CreER; shha^{ct/ct}* embryos treated with the vehicle or 4-HT. Severe cardiac edema was observed in 4-HT–treated embryos at 96 hpf (six abnormal in eight analyzed; right, arrowheads) but not in vehicle-treated embryos (zero abnormal in eight analyzed; left, arrowheads; *n* = 8 each; p<0.01, Fisher's exact test). (**D**) Immunofluorescence of heart sections obtained from vehicle- or 4-HT–treated *tcf21:CreER; shha^{ct/ct}* embryos. Insets, single-channel images of Mef2 immunofluorescence. Dotted yellow lines in insets depict the outline of the ventricle. (**E**) Quantification of Mef2+ nuclei from the sections obtained from the vehicle- or 4-HT–treated *tcf21:CreER; shha^{ct/ct}* embryos in D (*n* = 13 and 12). The data are presented as the mean ± SEM (***p<0.001, Mann–Whitney U test). (**F**) Semi-qRT-PCR analysis of epicardial marker gene expression in hearts dissected from *tcf21:CreER; shha^{ct/ct}* embryos treated with the vehicle (−) or 4-HT (+). (**G**) Immunofluorescence staining of heart sections obtained from vehicle- or 4-HT–treated *tcf21:CreER; shha^{ct/ct}* embryos. Raldh2 immunofluorescence was detected in *tcf21*:DsRed2+ epicardial cells in vehicle-treated embryos (left, arrowheads) but not in 4-HT-treated embryos (right, arrowheads). Bottom panels, single-channel images of Raldh2 immunofluorescence. (**H**) Semi-qRT-PCR analysis of the expression of myocardial growth factor genes in hearts dissected from *tcf21:CreER; shha^{ct/ct}* embryos treated with the vehicle (−) or 4-HT (+). (**I**) Semi-qRT-PCR analysis of *shha*-dependent myocardial growth factor genes in purified cardiomyocytes (CM) and epicardial cells (Epi) obtained from 96 hpf hearts. Single confocal sections are shown in D and G. ve, ventricle. Scale bar, 10 μm.

The following figure supplements are available for figure 4:

**Figure supplement 1.** *shha* expression during heart development and regeneration.

**Figure supplement 2.** Inversion rate measurement.

development (*Figure 4I*). Thus, epicardium-specific inactivation of *shha* expression revealed that Shha plays a crucial role in RA synthesis and mitogen expression in the epicardium during heart development in zebrafish.

## Epicardium-specific inactivation of *shha* expression during zebrafish heart regeneration

The zebrafish has been used for regeneration research for more than a decade, but the cKO approach has yet to be applied to these studies. To illustrate the utility of our cKO approach in zebrafish regeneration research, we inactivated epicardial *shha* expression in injured hearts and

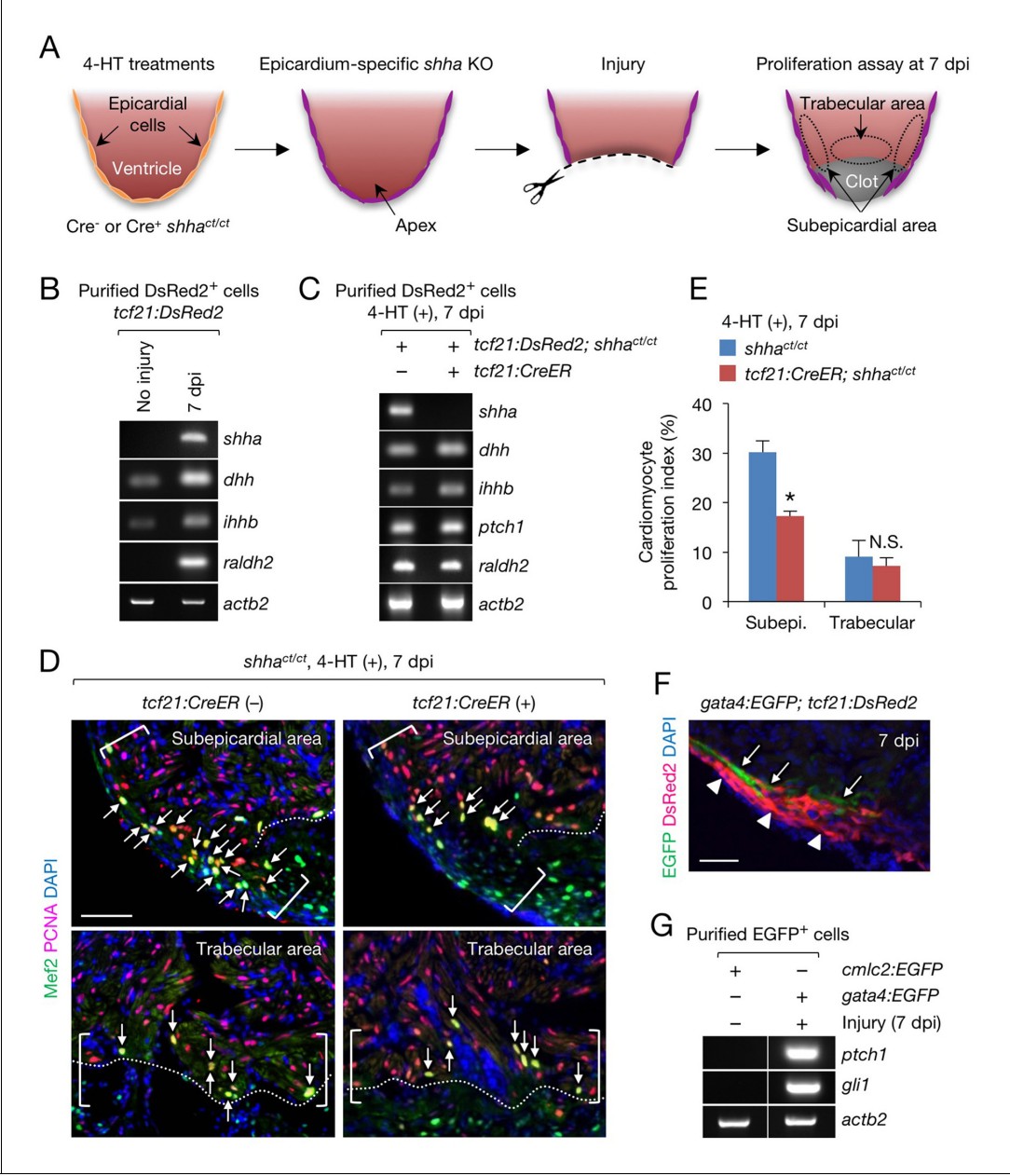

**Figure 5.** Epicardial *shha* expression promotes subepicardial cardiomyocyte proliferation during heart regeneration. (A) Schematic of the experiment. (B) Semi-qRT-PCR analysis of *shha* in purified *tcf21*:DsRed2⁺ epicardial cells obtained from uninjured and injured (7 dpi) *tcf21*:DsRed hearts. Injury was confirmed by the induction of *raldh2* expression. (C) Semi-qRT-PCR analysis of Hh pathway genes using purified *tcf21*:DsRed2⁺ epicardial cells obtained from 4-HT–treated 7 dpi *tcf21*:DsRed; *shha^{ct/ct}* (control, left) and *tcf21*:DsRed; *tcf21*:CreER; *shha^{ct/ct}* hearts (right). (D) Immunofluorescence images of the subepicardial (top) and trabecular areas (bottom) of heart sections obtained from 4-HT–treated 7 dpi *shha^{ct/ct}* (control) or *tcf21*:CreER; *shha^{ct/ct}* hearts. Brackets, subepicardial areas. Dotted lines, approximate amputation plane. Arrows indicate proliferating cardiomyocytes. (E) Quantification of cardiomyocyte proliferation in the subepicardial and trabecular areas of the heart sections obtained from 4-HT–treated 7 dpi *shha^{ct/ct}* (control) or *tcf21: CreER; shha^{ct/ct}* hearts shown in D (*n* = 6 each). The data are presented as the mean ± SEM (**p<0.01, Mann–Whitney U test). N.S., not significant (p=0.3367). (F) Image of heart sections obtained from 7 dpi *gata4:EGFP; tcf21;DsRed2* fish. Subepicardial cardiomyocytes (green, arrows) associate with epicardial cells (magenta, arrowheads). (G) Semi-qRT-PCR analysis of *shha* pathway genes using purified subepicardial cardiomyocytes obtained from 7 dpi *gata4:EGFP* ventricles. Cardiomyocytes purified from uninjured *cmlc2:EGFP* ventricles were used as negative controls. Scale bar, 50 μm.

The following figure supplements are available for figure 5:

**Figure supplement 1.** Assessment of spontaneous cassette inversion.

*Figure 5 continued on next page*

*Figure 5 continued*

**Figure supplement 2.** A redundant role for epicardial *shha* in epicardial migration and proliferation during heart regeneration.

investigated its effects on regeneration (*Figure 5A*). Studies using cyclopamine, which globally inhibits Hh signaling, indicated that epicardial Shha promotes myocardial proliferation (*Choi et al., 2013*) and epicardial regeneration in cultured injured hearts in vitro (*Wang et al., 2015a*). Consistent with the observation that the *shha* promoter is activated in regenerating epicardium (*Choi et al., 2013*), we detected *shha* expression in epicardial cells via in situ hybridization (*Figure 4—figure supplement 1D–F*) and semi-qRT-PCR in purified epicardial cells from *tcf21:DsRed2* hearts at 7 days postinjury (dpi) (*Figure 5B*). Next, we analyzed adult *tcf21:DsRed2; tcf21:CreER; shha^{ct/ct}* fish treated with 4-HT, which induced a nearly complete inversion of the gene trap cassette (*Figure 4—figure supplement 2E*). We used *tcf21:DsRed2; shha^{ct/ct}* fish treated with 4-HT as a negative control (CreER⁻) because a low level of cassette inversion was detected in *tcf21:CreER; shha^{ct/ct}* fish in adults, but not in embryos, in the absence of 4-HT (*Figure 5—figure supplement 1A–C*). Semi-qRT-PCR confirmed that *shha* expression was depleted in epicardial cells purified from injured hearts (*Figure 5C*). Interestingly, unlike the embryonic epi-KO heart (*Figure 4B and F*), *ptch1* and *raldh2* expression was unchanged in *shha* mutant adult hearts (*Figure 5C* and *Figure 5—figure supplement 2A and B*). Moreover, epicardial cells normally infiltrated the injury site (*Figure 5—figure supplement 2A–C*) and incorporated 5-ethynyl-2′-deoxyuridine (EdU), a DNA synthesis marker, in epi-KO hearts (*Figure 5—figure supplement 2C and D*). Consistent with the previous report (*Wang et al., 2015a*), we detected the expression of other Hh ligand genes, namely *dhh* and *ihhb*, in epicardial cells purified from injured hearts (*Figure 5B*) and epi-KO hearts (*Figure 5C*). These results suggest that redundant Hh proteins or non-epicardium-derived Shha plays a role in RA synthesis and epicardial regeneration in the adult zebrafish heart.

In contrast to epicardial cell proliferation, myocardial cell proliferation, defined as cardiomyocytes colabeled with immunofluorescence using anti-Mef2 and anti-proliferating cell nuclear antigen (PCNA) antibodies, was significantly decreased in the subepicardial area of injured epi-KO hearts (*Figure 5D and E*). However, myocardial cell proliferation was not reduced in the trabecular myocardium, an area distant from the epicardium (*Figure 5D and E*). To investigate the interplay between the epicardium and subepicardial myocardium, we crossed *tcf21:DsRed2* fish with *Tg(gata4:EGFP)* (*gata4:EGFP*) fish (*Heicklen-Klein and Evans, 2004*), as *gata4:EGFP* labels the subepicardial myocardium during cardiac growth (*Gupta et al., 2013*) and regeneration (*Kikuchi et al., 2010*). Immunofluorescence analysis of injured *tcf21:DsRed2; gata4:EGFP* hearts revealed that epicardial cells and subepicardial cardiomyocytes directly interact (*Figure 5F*), and semi-qRT-PCR analysis revealed that the expression of Hh target genes was strongly induced in *gata4*:EGFP⁺ subepicardial cardiomyocytes but not in control cardiomyocytes purified from uninjured *Tg(cmlc2:EGFP)* (*cmlc2:EGFP*) hearts (*Figure 5G*). These findings support the role of epicardial Shha in transmitting direct, short-range signals that promote the proliferation of adjacent cardiomyocytes during zebrafish heart regeneration.

## Discussion

In this study, we developed a streamlined method to efficiently generate conditional alleles in zebrafish using the invertible gene trap cassette *Zwitch*. We demonstrated that *Zwitch* can be inserted into a defined locus of the zebrafish genome via precise in vivo genome editing for inducible Cre-mediated gene disruption. In theory, the approach established in this study can be used to generate conditional alleles of any zebrafish gene, and it will expand the utility of zebrafish as a model organism.

Recently, TALEN-mediated HR was used to generate a floxed *kcnh6a* allele in zebrafish (*Hoshijima et al., 2016*). The loxP sites were provided by a donor plasmid that expressed the *α-crystallin:Venus* reporter gene, which was used to identify founders carrying the modified alleles with an overall germline transmission efficiency of 12% (*Hoshijima et al., 2016*). Using *Zwitch* in this study, we achieved a germline transmission efficiency of 59% and confirmed that 89% of alleles were in the correct non-mutagenic orientation (*Figure 1E*). The efficiency of the *Zwitch* method was likely

enhanced by screening for the LG marker at two different time points (7 and 45 dpf), as this excluded false positives resulting from unintegrated targeting vectors at 7 dpf (*Figure 1E*). Moreover, the ability of *Zwitch* to be inserted at any location in the target intron allowed us to screen for the TALEN pair associated with the greatest HR efficiency (*Figure 1B and C*). This feature may facilitate the efficient generation of conditional alleles in other genes, as a high DSB rate increases the rate of HR at target sites, thereby increasing the likelihood of successfully generating conditional alleles via genome editing (*Hoshijima et al., 2016*).

After pioneering studies demonstrated the editing of zebrafish genome with long dsDNA (*Bedell et al., 2012*; *Zu et al., 2013*), several studies reported the insertion of large DNA fragments at defined zebrafish gene loci via DSB-mediated HR (*Hisano et al., 2015*; *Hoshijima et al., 2016*; *Shin et al., 2014*; *Zu et al., 2013*) or homology-independent repair mechanisms (*Auer et al., 2014*; *Kimura et al., 2014*; *Li et al., 2015*). The efficiency of inserting the *Zwitch* construct may be increased via combination with these reported approaches. For example, a recent report suggested that asymmetric homology arm size, that is, 1 kb for one arm and 2 kb for the other arm, and the presence of a DSB in the shorter homology arm ensure efficient HR, thereby increasing the germline transmission rate of mutations (*Shin et al., 2014*). Designing homology arms following this approach may increase the targeting efficiency of the *Zwitch* construct. Recent studies using the CRISPR/Cas9 system reported that concurrent cleavage of the donor vector and target genome site induces efficient integration of a DNA fragment into a defined locus via homology-independent repair mechanisms (*Auer et al., 2014*; *Kimura et al., 2014*). Although we could not efficiently induce DSBs in the *shha* intronic sequence using the CRISPR/Cas9 system, this approach may be useful for targeting *Zwitch* into other genes.

We attempted to visualize mutant cells by connecting the TagRFP reporter gene to the splice acceptor site via a bicistronic 2A peptide sequence in *Zwitch* (*Figure 1A* and *Figure 1—figure supplement 1A and B*). We detected TagRFP expression in cells in the ganglion cell layer (GCL) in the retina (arrowheads, *Figure 2—figure supplement 3*), in which robust *shha* expression was previously reported (*Shkumatava et al., 2004*). However, the expression level was variable between GCL cells, compared with the uniform expression of endogenous *shha* mRNA in this tissue (*Figure 2—figure supplement 3*). Moreover, TagRFP expression was inconsistent in other tissues known to produce Shha at high levels, such as the floor plate and notochord. Given the transcription of *shha-2A-TagRFP* mRNA (*Figure 2C* and *Figure 2—figure supplement 2B*), the variability in the expression of TagRFP might be attributed, at least in part, to inefficient cleavage of the 2A peptide in the target cells. It is of interest to examine whether TagRFP expression would be enhanced in the target tissues by modifying the 2A sequence of *Zwitch* to include a glycine-serine-glycine spacer (GSG), which was recently found to significantly improve the cleavage efficiency of the 2A peptide (*Wang et al., 2015b*). We did not detect TagRFP expression in conditional *shha* mutant epicardial cells during development and regeneration (data not shown). We suspect that the failure to visualize *shha*-deficient epicardial cells expressing TagRFP was due to the low expression levels of endogenous *shha* in the epicardium during heart development and regeneration. Further studies are needed to improve the visualization of mutant cells for conducting mosaic studies using *Zwitch*.

Previously inaccessible aspects of organ morphogenesis can be addressed by combining conditional genetic analysis with established embryological, pharmacological, and transgenic manipulations in the zebrafish system. Moreover, the cKO approach will enhance the utility of zebrafish in elucidating the mechanisms underlying biological phenomena in adults that cannot be adequately investigated using a global KO approach. As an example, we used the *shha* gene trap line with an epicardium-specific inducible Cre driver and provided the first direct evidence that *shha* expression in the epicardium is required for the synthesis of RA and cardiomyocyte mitogens in the epicardium during heart development (*Figure 4G and H*). We further provided evidence that Shha may transmit a short-range signal to induce subepicardial muscle cell proliferation in the regenerating heart (*Figure 5D and E*), a mechanism that was not identified in studies of cyclopamine-mediated global Hh signaling inhibition (*Choi et al., 2013*). In situ hybridization analysis of embryo and adult hearts detected the expression of *shha* in *tcf21*:DsRed-negative cells (*Figure 4—figure supplement 1A–F*), suggesting that non-epicardial cells also produce Shha. Moreover, semi-qRT-PCR analysis detected the expression of *dhh* and *ihhb* in epicardial cells purified from epi-KO hearts (*Figure 5C*). Investigations of *shha* function using broader Cre driver lines and conditional inactivation of other Hh ligand genes will be needed in the future to definitively clarify the function of Shha during heart

development and regeneration. However, the results described in this study demonstrate that conditional genetic analysis is a feasible, effective approach to elucidate developmental and regenerative mechanisms in zebrafish, and it might also be useful for deciphering other complex phenomena such as immune responses, metabolism, and behavior.

In conclusion, we established a simple and efficient genome editing approach to engineer conditional alleles in zebrafish via HR-mediated *Zwitch* insertion. The tools and methods described in this study can be used to generate conditional alleles of other zebrafish genes, and they may be generally applicable to any experimental system in which CRISPR/TALEN gene editing is available.

## Materials and methods

### Zebrafish

The zebrafish used in this study were outcrossed from the Ekkwill (EK) background. All transgenic strains were analyzed as hemizygotes. The following published transgenic strains were used: *Tg (cmlc2:DsRed2)$^{pd15}$* (RRID:ZFIN_ZDB-ALT-110210-15) (*Kikuchi et al., 2010*), *Tg(gata5:EGFP)$^{pd25}$* (RRID:ZFIN_ZDB-ALT-110408-5) (*Kikuchi et al., 2011*), *Tg(tcf21:CreER)$^{pd42}$* (RRID:ZFIN_ZDB-ALT-110818-7) (*Kikuchi et al., 2011*), *Tg(gata4:EGFP)$^{ae1}$* (RRID:ZFIN_ZDB-ALT-051123-4) (*Heicklen-Klein and Evans, 2004*), and *Tg(cmlc2:EGFP)$^{f1}$* (RRID:ZFIN_ZDB-GENO-080403-2) (*Burns et al., 2005*). *Tg(ubb:iCRE-GFP)$^{vcc9}$* was generated by co-injecting pUbb-iCRE-GFP (*Figure 2—figure supplement 1A*) with I-*Sce*I into single cell-stage–embryos. Zebrafish were used for regeneration experiments at 4–12 months of age. The heart injury experiments were conducted as previously described (*Poss et al., 2002*). The animals were maintained at a density of 3–5 fish per liter, and clutch-mates of the appropriate genotypes were used as controls. The zebrafish husbandry procedures and all experiments were conducted in accordance with institutional and national animal ethics guidelines.

### Generation of the *shha*$^{ct}$ allele

The TALENs used to facilitate the insertion of *Zwitch* were designed using TAL Effector Nucleotide Targeter 2.0 software (*Doyle et al., 2012*) and constructed using the Golden Gate assembly method (*Cermak et al., 2011*; *Sakuma et al., 2013*). The TALENs were cloned into pCS2TAL3DDD and pCS2TAL3RRR vectors (provided by Dr. David Grunwald, University of Utah) (*Dahlem et al., 2012*). TALEN mRNAs were synthesized from linearized vectors using the mMESSAGE mMACHINE SP6 Transcription Kit (Thermo Fisher Scientific, Waltham, MA), and they were co-injected with pZwitch-shha-int1 into one-cell-stage embryos. The modified alleles were characterized using genomic PCR. The sequences of the PCR primers used to characterize the alleles are listed in *Supplementary file 1*. The offspring of a single founder were propagated and used in all subsequent experiments. The established conditional gene trap line *Tg(shha:Zwitch)$^{vcc8Gt}$* is referred to as *shha*$^{ct}$. A DIG DNA labeling and detection kit (Roche) was used for Southern blot analysis of genomic DNA isolated from adult *shha*$^{ct/+}$ fish and WT clutch-mates. The detection probe was generated using PCR with WT genomic DNA and the following primers:
5′-TTGTGGTTTACTCTATCAATCAACAGCCACAAGTGTTGTAGAGCT-3′ and
5′-AGTCTATTATACAAACGATATAGTCTAATGAAATAAATTGCAAAA-3′.
Probe labeling, hybridization, and detection were conducted according to the manufacturer's protocol.

### Genotyping

Embryos and adult *shha*$^{ct}$ fish were genotyped using genomic PCR. The primers are listed in *Supplementary file 1*. Embryos were generated by intercrossing *shha*$^{ct/+}$ fish, crossing *tcf21:CreER; shha*$^{ct/+}$ with *shha*$^{ct/+}$ fish, or crossing *tcf21:CreER; shha*$^{ct/+}$ with *tcf21:DsRed2; shha*$^{ct/+}$ fish. Embryos were genotyped individually by PCR with template DNAs prepared from single embryos as described subsequently (sample preparation for embryo genotyping and analysis). The WT allele was detected by PCR using F1 and R1 primers (*Figure 1B*). Amplification by F1/R1 primers was intervened in the conditional trap allele with the inserted *Zwitch* (4448 bp; *Figure 3—figure supplement 1A*). The conditional trap allele was detected by PCR using F2 and R2 primers for assessing germline transmission (*Figure 1D*) or F1 and R2 primers for genotyping (*Figure 3—figure supplement 1A*). The inverted, hence mutagenic, allele was detected by PCR using F4 and F5 primers (*Figure 2B*).

PCR using Cre-scr-F/R primers was performed to confirm *tcf21:CreER* and *ubb:Cre-GFP* transgenes (*Figure 3—figure supplements 1C* and *2B*). Examples of the genotyping results are shown in *Figure 3—figure supplements 1* and *2*.

Cre DNA and mRNA injection (*Figure 3A–C* and *Figure 4—figure supplement 2B and C*) or tamoxifen treatment (*Figure 4B–H*) was performed on embryos prior to genotyping PCR. Imaging of embryos (*Figures 3B* and *4C*) and screening of the *tcf21:DsRed2* transgene (*Figure 4G*) or GFP$^+$ embryos after Cre DNA injection (*Figure 3A–C*) were also performed on anesthetized live embryos prior to the PCR genotyping using an epifluorescence microscope.

## Sample preparation for embryo genotyping and analysis

For histologic analysis (*Figure 4D and G*), the upper body of embryos, including the heart, was separated from the lower body at the caudal end of the yolk sac extension using sharp forceps. The upper body was used for analysis, and the lower body was used for genotyping. The upper body of each embryo was transferred to an individual 1.5 ml tube containing 100 µl of 4% paraformaldehyde (PFA) and fixed at room temperature for 60 min, followed by rinsing with fix buffer (100 mM Na$_2$HPO$_4$ [pH 7.4], 4% sucrose, and 0.12 µM CaCl$_2$). The samples were stored in fix buffer at 4°C until embryo genotyping was performed. The lower body was collected into a PCR tube containing 50 µl of DNA extraction buffer (10 mM Tris-Cl [pH 8.0], 2 mM EDTA, 0.2% NP-40, and 200 µg/ml proteinase K) and used for genomic DNA extraction. Genomic DNA was extracted from the collected tissues by incubating the sample tubes at 50°C for 60 min, followed by proteinase inactivation at 95°C for 5 min. One microliter of the DNA solution was used as a PCR template, and genotyping PCR was conducted using a PrimeSTAR 1 GXL kit (Clontech, Mountain View, CA).

For qRT-PCR analysis (*Figure 3A*), the embryos were dissected similarly, but RNAlater (Thermo Fisher Scientific, Waltham, MA) was used for fixation. The upper body of each embryo was transferred to an individual 1.5 ml tube containing 100 µl of RNAlater, fixed at room temperature for 10 min, and stored at −80°C until embryo genotyping was performed. The lower body was used for PCR genotyping as described above. After genotyping, 10 pooled upper bodies of the same genotype were transferred to an individual 1.5-ml tube containing 1 ml of TRIzol (Invitrogen, Carlsbad, CA) and used for qRT-PCR analysis as described subsequently (RT-PCR).

For semi-qRT-PCR analysis (*Figure 4B,F,H*), the embryos were fixed with RNAlater at room temperature for 10 min, and the heart was dissected using sharp forceps. Each heart was transferred to an individual 1.5-ml tube containing 100 µl of RNAlater, and the samples were stored at −80°C until embryo genotyping was performed. The remaining body was collected into a PCR tube containing 50 µl of DNA extraction buffer and used for PCR genotyping as described above. After genotyping, 15 to 20 pooled hearts of the same genotype were transferred to an individual 1.5-ml tube containing 1 ml of TRIzol and used for semi-qRT-PCR analysis as described subsequently (RT-PCR).

## Tamoxifen treatment

Zebrafish were treated with 4-HT as previously described (*Kikuchi et al., 2011*). Briefly, at 24 hpf, the embryos were placed in embryo medium supplemented with 5 µM 4-HT produced from a 1 mM 4-HT stock solution in 100% ethanol. After 24 hr, the embryos were transferred to fresh 4-HT-containing medium for an additional 24 hr.

Adult zebrafish were placed in a small beaker of aquarium water supplemented with 5 µM 4-HT for 12 hr and subsequently transferred to fresh 4-HT-containing medium for an additional 12 hr. Then, the fish were rinsed with fresh aquarium water and returned to the recirculating water system. Vehicle- or 4-HT-treated fish were used for regeneration experiments or epicardial cell isolation 3 days after the treatment.

## Inversion rate measurement

We estimated the inversion efficiency using a PCR-based approach as follows. First, a DNA segment of the non-mutagenic or the mutagenic allele was amplified by PCR using F7 and R7 primers (*Figure 4—figure supplement 2A*). The PCR products were purified using a Wizard SV PCR and Gel Purification kit (Promega, Madison, WI) and digested with *BglI*. Via this digestion, the PCR products specific to the mutagenic allele (M1 and M2 bands; *Figure 4—figure supplement 2B*) were separated from those specific to the non-mutagenic allele upon gel electrophoresis (N1 and N2 bands;

*Figure 4—figure supplement 2B*). The intensity of each band was quantified using ImageJ software, and the value was normalized by the size (bp) of the band, providing the relative mole quantity (q) of DNA molecules in each band. The inversion efficiency was obtained as the percentage of the ratio of the sum of the relative mole quantity of M1 and M2 ($qM1 + qM2$) to that of all DNA bands ($qM1 + qM2 + qN1 + qN2$). Examples of the estimated inversion rate and its correlation to *shha* expression levels are shown in *Figure 4—figure supplement 2C*.

## Plasmids

### pZwitch
The LG cassette was PCR amplified from an *α-crystallin:EGFP* cassette (*Kikuchi et al., 2011*), and the PCR product was inserted between two FRT sites of pL451 (*Liu et al., 2003*) (provided by Dr. Stephan Creekmore, NCI-Frederick). The resulting vector was referred to as pL451-FRT-LG-FRT. Tandem LoxP-Lox5171 sites, the splice acceptor (SA) sequence of pFT1 (provided by Dr. Wenbiao Chen, Vanderbilt University) (*Ni et al., 2012*), and the P2A sequence (SA-P2A) were synthesized using Ultramer Oligo Synthesis by IDT Technologies. TagRFP cDNA was PCR-amplified from pTagRFP-C (Evrogen, Moscow, Russia), and 5× BGHpA repeats were derived from pFT1 via restriction enzyme digestion. The components were inserted into pL451-FRT-LG-FRT via restriction enzyme digestion, and the resulting construct was referred to as pZwitch. As this version was designed for +1 reading frame, it was designated pZwitch+1. To use different reading frames, we created pZwitch+2 and pZwitch+3 using SA-P2A oligonucleotides with the addition of one or five base pairs upstream of the P2A sequence (*Figure 1—figure supplement 1B*).

### pZwitch-shha-int1
The LA and RA of the homology sequences were amplified from genomic DNA isolated from adult EK zebrafish using the following primers:
1) LA amplification primers:
5′-TTGCTAGCCGAGCATAATTTTTATTAGGCTGTTTTGAACGTGCCTCTGTTAAA-3′ (LA-F) and
5′-TTACGCGTAGTCTATTATACAAACGATATAGTCTAATGAAATAAATTGCAAAA-3′ (LA-R).
2) RA amplification primers:
5′-TTGAATTCCTAGAAACACAATGAAGTGAACTGAACAGTTTCATACATTCAGAT-3′ and 5′-TTCTCGAGTCAGCCATGTCAGTCAGTTCTGAGCAACCATCTCAGGAATCAGAG-3′.
The LA PCR product was digested with *Nhe*I and *Mlu*I, and the RA PCR product was digested with *EcoR*I and *Xho*I. The products were cloned into the corresponding restriction enzyme sites in pZwitch.

### pUbb-iCRE-GFP
Approximately 4 kb of the 5′ upstream sequence of *ubb* was amplified using the following primers:
5′-GGGGGAAACGGAGCACCCACTCAAATGCAGGGAGAACATGCAAAC-3′ and
5′-CTGTAAACAAATTCAAAGTAAGATTAGCAATCACACATTTCCATT-3′.
Purified DNA from a BAC clone (CH211-202A12; BACPAC Resources Center, CHORI, CA) encoding *ubb* was used as the PCR template. Codon-improved Cre (iCRE) cDNA was PCR amplified from pDIRE (Addgene plasmid #26745; provided by Dr. Rolf Zeller, University of Basel) (*Osterwalder et al., 2010*), and EGFP cDNA was PCR-amplified using synthetic oligonucleotides with the P2A sequence as the 5′ primer. The components were assembled into pL451 using restriction enzyme digest cloning, and synthetic oligonucleotides with I-*Sce*I (NEB, Ipswich, MA) sites were inserted at the 5′ and 3′ ends of the construct.

### pCS2-FLPo
Codon-optimized FLP (FLPo) was PCR amplified from pDIRE and inserted into pCS2+ using restriction enzyme digest cloning. To remove the LG tag, Flp mRNA was synthesized from linearized pCS2-FLPo using the mMESSAGE mMACHINE SP6 kit, and the resulting mRNA was injected into single-cell embryos.

### pCS2-iCRE-BFP

iCRE cDNA was PCR amplified from pDIRE and TagBFP cDNA was PCR amplified using synthetic oligonucleotides with the P2A sequence as the 5′ primer. The components were inserted into pCS2+ using restriction enzyme digest cloning. Cre mRNA was synthesized from linearized pCS2-iCRE-BFP using the mMESSAGE mMACHINE SP6 kit, and the resulting mRNA was injected into single-cell embryos.

## Cre DNA and mRNA injection

To increase the mosaicism of Cre expression, pUbb-iCRE-GFP DNA was co-injected with I-*Sce*I into one-cell–stage embryos as previously described (*Thermes et al., 2002*). The injected embryos were examined at 3 dpf using an MVX10 microscope (Olympus, Tokyo, Japan). Embryos that expressed EGFP on approximately >80% of the total body surface area were selected for qRT-PCR and phenotypic analysis (*Figure 3A–C*). The embryos expressing EGFP at these levels exhibited the *Zwitch* inversion at more than 90% efficiency (*Figure 4—figure supplement 2C*).

## Flow cytometry

Embryonic epicardial cells and cardiomyocytes were isolated using a previously published protocol (*Burns and MacRae, 2006*). We extracted hearts from approximately 500 transgenic embryos harboring *tcf21:DsRed2; gata5:EGFP* (*Kikuchi et al., 2011*) or *cmlc2:DsRed2* (*Kikuchi et al., 2010*). In *tcf21:DsRed2; gata5:EGFP* embryos, EGFP and DsRed2 colocalization was restricted to the epicardium (*Kikuchi et al., 2011*), and this effect enabled us to isolate highly purified epicardial cells (*Figure 4A*). The extracted hearts were placed in a Petri dish containing ice-cold 1× Hanks' Balanced Salt Solution (HBSS) and examined using an MVX10 microscope. To prepare single-cell suspensions, hearts expressing the fluorescent reporter were collected with a pipette and transferred to a 1.5-ml tube containing 1× HBSS with 1 mg/ml collagenase type 2 (Worthington Biochemical, Lakewood, NJ). The samples were incubated for 40 min at room temperature and mixed by gentle pipetting every 10 min. The dissociated cells were washed and resuspended in ice-cold 1× HBSS containing 5% fetal bovine serum (FBS).

To isolate adult epicardial cells and cardiomyocytes, ventricles were extracted from adult zebrafish and minced using sharp forceps in ice-cold 1× HBSS. To obtain single-cell suspensions, minced ventricle tissues were transferred to a 1.5-ml tube containing 1× HBSS with 1 mg/ml collagenase type 2. The samples were incubated for 40 min at room temperature and mixed by gentle pipetting every 10 min. Dissociated cells were washed and suspended in ice-cold 1× HBSS containing 5% FBS.

The cells were sorted using a FACSAria IIU 16 platform with the purity mode (BD Biosciences, Franklin Lakes, NJ). The cells were directly collected into a 1.5-ml tube containing 1 ml of TRIzol reagent and subsequently used for RT-PCR analysis. Dead cells, defined as those stained with DAPI, were excluded from the cell sorting experiments.

## RT-PCR

The qRT-PCR analysis used 10 upper bodies of the same genotype (*Figure 3A*), and the semi-qRT-PCR analysis used 15–20 hearts of the same genotype (*Figure 4B,F,H*). Total RNA was extracted using TRIzol, and cDNA was subsequently synthesized using a Transcriptor First-Strand cDNA Synthesis Kit (Roche, Basel, Switzerland). qRT-PCR was conducted using a LightCycler 480 system (Roche, Basel, Switzerland). For semi-qRT-PCR, genes of interest were amplified using a PrimeSTAR GXL kit. cDNA levels were normalized to *actb2/β-actin2* levels in both the qRT-PCR and semi-qRT-PCR experiments. The primers used for semi-qRT-PCR and qRT-PCR analysis are listed in *Supplementary file 1*.

## Histological analysis and imaging

In situ hybridization for *raldh2* and immunofluorescence staining assays were conducted as previously described (*Kikuchi et al., 2011*), and the results were imaged using a LeicaDM400 B LED microscope with an MC170 HD camera (Leica Microsystems, Wetzlar, Germany) or a Zeiss AXIO imager M1 microscope (Carl Zeiss AG, Oberkochen, Germany). Confocal images were captured using a Zeiss LSM 710 confocal microscope (Carl Zeiss AG, Oberkochen, Germany). In situ hybridization for *shha* was performed using RNAscope Reagent kits (Advanced Cell Diagnostics, Newark,

CA), and the results were imaged using a LeicaDM400 B LED microscope and a Zeiss LSM 710 confocal microscope. EdU (8 mM) was intraperitoneally injected at 4, 5, and 6 dpi, and EdU incorporation was detected using a Click-iT EdU Alexa 647 Imaging Kit (Invitrogen, Carlsbad, CA).

The following primary antibodies were used: rabbit anti-DsRed (Invitrogen, Carlsbad, CA), rabbit anti-tRFP (Evrogen, Moscow, Russia), chicken anti-GFP (Abcam, Cambridge, United Kingdom), rabbit anti-Mef2 (Santa Cruz Biotechnology, Dallas, TX), mouse anti-myosin heavy chain (Clone F59; Developmental Studies Hybridoma Bank, Iowa City, IA), mouse anti-PCNA (Sigma-Aldrich, St. Louis, MO), and rabbit anti-zf Raldh2 (Abmart, Berkeley Heights, NJ). The following secondary antibodies were used: Alexa Fluor 488 donkey anti-mouse IgG(H + L), Alexa Fluor 488 donkey anti-rabbit IgG(H + L), Alexa Fluor 555 donkey anti-mouse IgG(H + L), and Alexa 24 Fluor 555 donkey anti-rabbit IgG(H + L) (Invitrogen, Carlsbad, CA).

## Quantification

The number of cardiomyocytes in embryonic hearts was determined using single confocal sections (1 μm thick) of ventricles that were captured using a Zeiss LSM 710 confocal microscope with a 25× objective. The number of $Mef2^+DAPI^+$ nuclei in ventricles stained with anti-MHC was manually quantified using ImageJ software. The number of cardiomyocytes was determined by calculating the average number of $Mef2^+DAPI^+$ nuclei in three sections from each heart.

To quantify subepicardial myocardial cell proliferation, images of the injury border were captured using a Zeiss AXIO imager M1 microscope with a 10× objective (967 × 267 pixels). The subepicardial region was defined as the area within approximately 50 μm of the epicardium, and the trabecular area was defined as the area encompassing the injury, excluding the area used to quantify subepicardial myocardial proliferation. The numbers of $Mef2^+$ and $Mef2^+PCNA^+$ cells in the subepicardial and trabecular regions were manually quantified using ImageJ software. The cardiomyocyte proliferation index was defined as the percentage of $Mef2^+PCNA^+$ cells in three selected sections from each heart.

To quantify epicardial cell proliferation, images of the wound area and the uninjured right ventricular wall were captured at 7 dpi using a Zeiss AXIO imager M1 microscope with a 10× objective (967 × 267 pixels). The numbers of $tcf21$:$DsRed2^+DAPI^+$ nuclei and $tcf21$:$DsRed2^+EdU^+$ nuclei were manually determined using ImageJ software. The epicardial cell proliferation index was defined as the percentage of $tcf21$:$DsRed2^+EdU^+$ nuclei in three selected sections from each heart.

## Data collection and statistics

Sample sizes were determined in previous publications, and experiment types and are indicated in the legends. Representative results of at least two repeated experiments are shown. All experiments were performed with at least five biological replicates. Individual fish or pooled individual embryos were used as biological replicates. No data were excluded unless the animal died during the procedure. The Mann–Whitney U test and Fisher's exact test were used. The statistical methods used and p values are indicated in the legends. Raw quantification data are available in *Supplementary file 2*.

## Targeting *Zwitch* into other genes

### Identification of target introns

Open the Ensembl genome browser (http://www.ensembl.org/Danio_rerio/Info/Index) and search for the target gene using the zebrafish genome. On the target gene page, choose 'Transcript ID' for each transcript variant. In the 'Transcript-based displays panel' on this page, select 'Exons' under 'Sequence' to obtain sequence information and identify the translated exon containing the ATG start codon. Repeat this step for all variants and identify the ATG exon included in all potential transcript variants. Any introns located downstream of this ATG exon can be used for *Zwitch* insertion. The exon usage may need to be experimentally confirmed using target cells and/or tissue samples.

### Designing candidate TALENs

In the Ensembl page of the target gene, click 'Configure this page' and change the number at 'Intron base pairs to show at splice sites' to display the entire sequence of the intron. Open TAL Effector Nucleotide Targeter 2.0 software (https://tale-nt.cac.cornell.edu/node/add/talen) and paste the target intron sequence in the FASTA format in the 'Sequence' box. Select 'Provide Custom

Spacer/RVD length' and search TALENs with the following parameters: Minimum Spacer Length, 13; Maximum Spacer Length, 20; Minimum Repeat Array Length, 15; Maximum Repeat Array Length 15; G Substitute, NH; Filter Options, Show TALEN pairs (hide redundant TALENs); Streubel et al. guidelines, Off; Count Predicted Targets in a Genome/Proteome, Danio rerio (genome), Scoring Matrix, Doyle et al; and Upstream Base, T only. Select several TALENs that have no or the smallest number of off-targets, contain a restriction enzyme site in its spacer sequence, and bind to the intronic sequence at least 100 bp from the exon to avoid interfering splicing machinery.

## Assembly and synthesis of TALENs

Assemble TALENs using a Golden Gate TALEN and TAL Effector Kit 2.0 (Addgene kit # 1000000024) (*Cermak et al., 2011*). Clone the assembled TALENs into pCS2TAL3DDD and pCS2TAL3RRR vectors (provided by Dr. David Grunwald, University of Utah) (*Dahlem et al., 2012*), and synthesize mRNAs using the mMESSAGE mMACHINE SP6 Transcription Kit.

## Efficiency test of TALENs

Inject approximately 50 pg of TALEN mRNAs into one-cell–stage embryos and purify genomic DNA after 48 hr from 10 to 20 pooled embryos using a Wizard Genomic DNA Purification Kit (Promega, Madison, WI). Also prepare genomic DNA from embryos that were not injected with TALENs as a negative control for the next measurement step. Use a Restriction Fragment Length Polymorphism (RFLP) assay and measure TALEN activities as described previously (*Ma et al., 2013*). Other assays such as T7 Endonuclease I cleavage assay or high-resolution melt curve analysis can also be used, but we prefer using the RFLP assay because it is inexpensive and reasonably accurate in determining TALEN activities. Select the TALEN pairs displayed the highest efficiency in inducing DSBs for the subsequent steps.

## Cloning of homology arms

Amplify 500–1000 bp DNA segment of 5′ upstream and 3′ downstream regions to the spacer sequence of the identified TALEN binding site using a PrimeSTAR GXL PCR Kit. Use primers including *Nhe*I or *Mlu*I sites at their 5′ ends to amplify the upstream DNA segment. Similarly, use primers including recognition sites for *Apa*I, *Xho*I, *Cla*I, *Eco*RV, or *Eco*RI at their 5′ end to amplify the downstream DNA segment. Insert the upstream DNA segment into the *Nhe*I and *Mlu*I sites of pZwitch as the left homology arm, and the downstream DNA segment into *Apa*I, *Xho*I, *Cla*I, *Eco*RV, and *Eco*RI sites as the right homology arm via restriction enzyme cloning. Purify the DNA of the resulting pZwitch construct and perform DNA sequencing of the cloned homology arms using the following primers:
i) For the left homology arm,
5′-GCGGATAACAATTTCACACAGG-3′ (M13R) and
5′- TGTGGACCACAGACTGGAAAGAA-3′ (pBS-SK-R); ii) For the right homology arm,
5′-GTAAAACGACGGCCAGT-3′ (M13F) and
5′-AGTTTGAATAACTACATATGCATAGG-3′ (Cryaa-5R).

## Injection of *Zwitch* construct

Obtain genomic DNA by tail fin clipping from the clutch-mate fish of the fish used for homology arm cloning. Perform DNA sequencing and identify several male and female fish whose DNA sequence matches the cloned homology arms. Use these fish to produce embryos for the injection of the pZwtich construct. Inject approximately 50 pg each of pZwtich construct DNA and TALEN mRNAs into one-cell–stage embryos.

## Prescreening of potential founder fish

Screen LG expression in injected embryos between 4 and 7 dpf and transfer LG+ embryos to aquarium tanks. Re-screen LG expression between 30 and 45 dpf and select fish maintaining LG expression uniformly in the eye. Do not select fish exhibiting punctate LG expression, as these fish often do not carry the integrated gene trap cassette (data not shown).

## Identification of founder fish

Outcross adult LG⁺ fish with WT fish and assess LG expression in the offspring. It is critical to perform genomic PCR diagnosis for all LG⁺ F1 fish to confirm the correct insertion of *Zwitch*. To validate insertion at the 5′ end of the target site, perform PCR using a forward primer binding to DNA sequences outside of the left homology arm and the following primer as a reverse primer:
5′-GTGGGGCAGGACAGCAAGGGGGAGGATTGGGAAGACAATAGCAGG-3′ (R2)
To validate the insertion at the 3′ end of the target site, perform PCR using a reverse primer binding to DNA sequences outside of the right homology arm and the following primer as a forward primer:
5′-ATTTACCAATGCAAGTTTCTCTGAAGATCGTTTATTGCCATATTA-3′ (F3)
Once the correct insertion of *Zwitch* is confirmed, propagate LG⁺ F1 fish for experiments.

## Acknowledgements

We thank K Poss for help in establishing the laboratory. We thank D Grunwald, S Creekmore, W Chen, and R Zeller for plasmids; S Malik and M Nakayama for technical assistance; C Jenkin, J Martin, and K Brennan for zebrafish care; Garvan Flow for cell sorting; and R Harvey and D Hesselson for comments on the manuscript. KS received postdoctoral fellowships for research abroad from the Japan Society for the Promotion of Science. KK was an ARC Future Fellow (FT110100836). This work was supported by grants from NHMRC (1032522 and 1046469).

# Additional information

### Funding

| Funder | Grant reference number | Author |
|---|---|---|
| National Health and Medical Research Council | APP1032522 | Kazu Kikuchi |
| Australian Research Council | FT110100836 | Kazu Kikuchi |
| National Health and Medical Research Council | APP1046469 | Kazu Kikuchi |
| Japan Society for the Promotion of Science | Postdoctoral Fellowship for Research Abroad | Kotaro Sugimoto |

The funders had no role in study design, data collection and interpretation, or the decision to submit the work for publication.

### Author contributions

KS, Conceptualization, Data curation, Formal analysis, Investigation, Project administration, Writing—review and editing; SPH, DZS, Data curation, Formal analysis, Writing—review and editing; KK, Conceptualization, Data curation, Formal analysis, Project administration, Supervision, Writing—original draft, Writing—review and editing, Funding acquisition

### Author ORCIDs

Kazu Kikuchi, http://orcid.org/0000-0002-4681-4275

### Ethics

Animal experimentation: This study was performed in strict accordance with the Australian code for the care and use of animals for scientific purposes. All of the animals were handled according to approved institutional animal care and use committee protocols of the Victor Chang Cardiac Research Institute. The protocol was approved by the Garvan Institute of Medical Research/St Vincent's Hospital Animal Ethics Committee (Permit Number: 14-18). All surgery was performed under tricaine anesthesia, and every effort was made to minimize suffering.

## Additional files

**Supplementary files**
• Supplementary file 1. Sequences of PCR primers used in this study.
• Supplementary file 2. Raw quantification data and statistical tests from experiments in this study.

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
