## [Decision Letter]

Thank you for submitting your article "Dissection of zebrafish *shha* function using site-specific targeting with a Cre-dependent genetic switch" for consideration by *eLife*. Your article has been favorably evaluated by K VijayRaghavan (Senior Editor) and four reviewers, one of whom, Alejandro Sánchez Alvarado (Reviewer #1), is a member of our Board of Reviewing Editors.

The reviewers have discussed the reviews with one another and the Reviewing Editor has drafted this decision to help you prepare a revised submission.

Summary:

In this manuscript, Kikuchi and colleagues generated a novel, invertible gene trap cassette, termed *Zwitch*, which they use as a part of a methodology for conditional gene abrogation in zebrafish. In the *Zwitch*construct they included a Lens GFP marker for determining transgene incorporation. Overall, the reviewers agree that there is potentially great significance to the application of the inversible gene trap cassettes that can be integrated site-specifically as described by the authors. However, all reviewers also expressed concerns regarding the biological findings reported. As such, we would like to recommend that the authors refocus their efforts in the requested revision to expand on the details of the methods so as to make *Zwitch* more accessible to the interested community.

Essential revisions:

The following points need to be fully and satisfactorily addressed by the authors before the paper can be considered for publication in *eLife*:

1) Throughout the manuscript, the authors did not directly show the efficiency of inversion in the target tissue. Although the gene expression data is supportive of efficient inversion of the gene trap, directly measured inversion rate is also important.

2) Results. "Epicardium-specific *shha* deletion during zebrafish heart development" and "To demonstrate the utility of *Zwitch*, we investigated the functional consequences of *shha* deletion in the epicardium, the mesothelial layer covering the heart." The word "deletion" here is not accurate as *Zwitch* only inactivates gene expression and is not able to generate a real deletion for the *shha* locus. This may also explain why not all embryos develop severe phenotypes.

3) Subsection “Epicardium-specific *shha* deletion during zebrafish heart regeneration”, end of first paragraph. Authors should show the expression pattern of *shha* during heart development and regeneration to rule out or confirm the possibility that non-epicardium-derived *shha* is present. Importantly, other broad drivers should be used to generate conditional knockout to see if stronger phenotypes can be detected during heart regeneration. If redundant Hh proteins are present, what are they? Where are they being expressed? How do you confirm weak phenotype are caused by redundancy but not because authors did not find the right one?

4) The authors attribute the normal epicardial gene expression during heart regeneration in epicardial *shha* deficient adults to redundant expression of other Hh genes. It would be interesting to determine whether there is any compensatory increase of the expression of these genes in the mutant heart.

5) The inability of detecting tagRFP is somewhat disappointing. I agree that low level of expression of the endogenous gene in epicardium may very well be a contributing factor. However, *shha* is highly expressed in the floor plate and notochord during development. It is unclear if the authors detected tagRFP signal in developing embryos after global inversion by Cre mRNA injection.

6) PCRs unfortunately do not include a positive control proving that samples that show no amplification contained DNA. For example, in 1F and 2B, C, E primers amplifying the wild-type shh locus or cDNA should be included.

7) Figure 3: it's unclear how the authors identified ct/ct embryos. Methods indicate that they did not raise homozygous adults, which then were incrossed. Rather it seems the experiments were done on individual embryos genotyped from an incross of ct/+ fish? Please clarify and make this also clear in the manuscript text. If the latter: an example of a genotyping PCR should be shown that can distinguish ct/+ from ct/ct embryos with and without Cre.

8) To prove that the inverted ct allele is a loss-of-function allele of shh, the authors should establish a non-mosaic line of heterozygous fish containing the mutagenic allele and analyze the phenotype of embryos derived from incrosses of these and quantify *shha* RNA expression (protein quantification is probably not possible due to non-availability of anti-Shha antibodies?)

9) 4C: sample size and frequency of the edema phenotype needs to be given.

10) Is the heart edema and reduced CM-number phenotype also observed in non-conditional *shha* mutants? If not (I could not find reports describing it), the authors should present a hypothesis why this would only be seen in the conditional mutant.

11) This article aims to provide a novel construct for using Cre-mediated recombination to conditionally ablate genes in zebrafish. Overall, the ability to efficiently introduce targeted mutations would be an important advance for functional work in zebrafish. The utility of this strategy seems contingent on the efficiency of homologous recombination. The manuscript would benefit by including an additional description of how the efficiency of their strategy compares with other published approaches. Is this really any better than what is out there?

12) For a methods paper, little information is given on the approach used for designing the left arm and right arm homology sequences and the TALENs. For a general audience, this needs to be expanded. Additionally, the Materials and methods should be expanded to include all details (including relevant product numbers) for vector construction and the specific steps that would be required for using the *Zwitch* construct to target other genes, such that anyone with reasonable molecular biology skills could follow step-by-step.

---

## [Author Response]

*Essential revisions:*

*The following points need to be fully and satisfactorily addressed by the authors before the paper can be considered for publication in eLife:*

*1) Throughout the manuscript, the authors did not directly show the efficiency of inversion in the target tissue. Although the gene expression data is supportive of efficient inversion of the gene trap, directly measured inversion rate is also important.*

We measured the inversion rate in the Cre-DNA–injected embryos used for the experiments in Figure 3, as mentioned in the subsection “Cre DNA injection” of the Materials and methods in the original manuscript. We used a PCR-based approach for this measurement, as it was difficult to prepare an adequate quantity of genomic DNA from embryos for Southern blot analysis. We have included a new subsection titled “Inversion rate measurement” in the Materials and methods in the revised manuscript and described how the inversion rate of the gene trap cassette was measured in embryos and epicardial cells. We presented the data in Figure 4—figure supplement 2.

We described the data in the revised manuscript as follows:

“We inactivated *shha* expression […], which induced a nearly complete inversion of the gene trap cassette (Figure 4—figure supplement 2).”

“Next, we analyzed adult *tcf21:DsRed2; tcf21:CreER; shha^ct/ct^*fish treated with 4-HT, which induced a nearly complete inversion of the gene trap cassette (Figure 4—figure supplement 2).”

*2) Results. "Epicardium-specific shha deletion during zebrafish heart development" and "To demonstrate the utility of Zwitch, we investigated the functional consequences of shha deletion in the epicardium, the mesothelial layer covering the heart." The word "deletion" here is not accurate as Zwitch only inactivates gene expression and is not able to generate a real deletion for the shha locus. This may also explain why not all embryos develop severe phenotypes.*

The reviewer's comment is correct. We changed “*shha* deletion” to “inactivation of *shha* expression” in the indicated sentences and wherever applicable in the revised manuscript as follows:

“Epicardium-specific inactivation of *shha* expression during zebrafish heart development”.

“To demonstrate the utility of *Zwitch*, we investigated the functional consequences of inactivation of *shha* expression in the epicardium…”.

“We induced epicardium-specific inactivation of *shha* expression…”. “Thus, epicardium-specific inactivation of *shha* expression…”. “Epicardium-specific inactivation of *shha* expression during zebrafish heart regeneration”.

“Figure 4. Epicardium-specific inactivation of *shha* expression during heart development”.

*3) Subsection “Epicardium-specific shha deletion during zebrafish heart regeneration”, end of first paragraph. Authors should show the expression pattern of shha during heart development and regeneration to rule out or confirm the possibility that non-epicardium-derived shha is present. Importantly, other broad drivers should be used to generate conditional knockout to see if stronger phenotypes can be detected during heart regeneration. If redundant Hh proteins are present, what are they? Where are they being expressed? How do you confirm weak phenotype are caused by redundancy but not because authors did not find the right one?*

To examine whether non-epicardium-derived *shha* is present during heart development and regeneration, we performed in situ hybridization analysis of *shha* and presented the result in Figure 4—figure supplement 1.

We used *tcf21:DsRed2* transgenic reporter zebrafish to confirm whether *shha* mRNA expression overlap with *tcf21*^+^ epicardial cells. In the developing heart, we detected *shha* mRNA expression (arrowheads) in the ventricle (ve), and confirmed that some expression in the ventricle was colocalized with *tcf21*-driven DsRed2 (arrows, Figure 4—figure supplement 1). We also detected *shha* mRNA expression in the bulbus arteriosus (ba) but did not observe colocalization of *tcf21*-driven DsRed2 with these *shha* mRNA signals (arrows, Figure 4—figure supplement 1). These signals may arise from non-epicardial cells. The expression of *shha* mRNA was also detected in non-cardiac tissues such as the skin and the epithelium of the esophagus the epithelium of the esophagus (es; Figure 4—figure supplement 1).

In the regenerating heart, we detected the majority of *shha* mRNA signals in the subepicardial area of the ventricle (bracket, Figure 4—figure supplement 1) and confirmed these signals largely overlapped with *tcf21*-driven DsRed2 (arrows, Figure 4—figure supplement 1). We also identified a few cells expressing *shha* mRNA that were not labeled with DsRed2 (arrow, Figure 4—figure supplement 1), suggesting that Shha may be produced at low level by non-epicardial cells during regeneration.

We agree that additional Cre driver lines should be used to delineate *shha* functions in other tissues, however importing such Cre driver lines and analyzing adult regeneration in the homozygous mutant background could not be completed within the given period for this revision. While our data does not rule out a role for extra-epicardial *shha* or the redundant action of other Hh ligands, the major focus of our study is to establish a genetic method facilitating an inducible, tissue-specific gene knockout analysis in zebrafish and demonstrate its utility in embryo and adult zebrafish. Further investigation of Shha function during regeneration using different Cre driver lines is important, but it will be a focus of a future study.

A previous study reported that the expression of Hh ligand genes, *dhh* and *ihhb*, is upregulated in injured zebrafish hearts (Wang J, et al. Nature 2015). Consistent with this result, our new semi-qRT-PCR result revealed the upregulation of *dhh* and *ihhb* expression in purified epicardial cells after injury (Figure 5). The expression of these genes was unchanged after the inactivation of *shha* expression (Figure 5). Figure 5 have been modified to include these results in the revised manuscript. The other Hh ligand genes *shhb* and *ihha* were undetectable, as previously reported (Wang J, et al. Nature 2015). Thus, the redundant Hh proteins present in injured zebrafish hearts are likely Dhh and Ihhb, and these ligands are expressed at least in the epicardium.

Among the Hh ligand genes expressed in the zebrafish heart, Shha is the ligand most strongly upregulated during regeneration (Wang J, et al. Nature 2015) (Figure 5). Therefore, we focused on Shha as the major Hh ligand in heart regeneration. Future studies employing conditional and combinatorial inhibition of Hh ligands will be necessary to definitively determine the relative contribution of each ligand.

We described the new results in the revised manuscript as follows:

“In situ hybridization analysis also detected *shha* expression in the epicardium of the ventricle (Figure 4—figure supplement 1).”

“…we detected *shha* expression in epicardial cells via in situ hybridization (Figure 4—figure supplement 1)…”.

“Consistent with the previous report (Wang et al., 2015a), we detected the expression of other Hh ligand genes, namely *dhh* and *ihhb*, in epicardial cells purified from injured hearts (Figure 5) and epi-KO hearts (Figure 5).”

We discussed the new results in the revised manuscript as follows:

“In situ hybridization analysis of embryo and adult hearts detected the expression of *shha* in *tcf21*:DsRed-negative cells (Figure 4—figure supplement 1), suggesting that non-epicardial cells also produce Shha. […] Investigations of *shha* function using broader Cre driver lines and conditional inactivation of other Hh ligand genes will be needed in the future to definitively clarify the function of Shha during heart development and regeneration.”

*4) The authors attribute the normal epicardial gene expression during heart regeneration in epicardial shha deficient adults to redundant expression of other Hh genes. It would be interesting to determine whether there is any compensatory increase of the expression of these genes in the mutant heart.*

We analyzed the expression of the Hh ligand genes *shhb, dhh, ihha*, and *ihhb* using 4-HT–treated injured *shha^ct/ct^*and *tcf21:CreER; shha^ct/ct^*heart samples via qRT-PCR. *shhb* and *ihha* expression was not detectable, consistent with the result described in a previous study (Wang J, et al. Nature 2015). The expression of *dhh* and *ihhb* was detected in the injured heart, but their expression was not significantly changed by the inactivation of *shha* expression in the epicardium. As detailed in the response to comment #3, we also examined the expression of *dhh* and *ihhb* in purified epicardial cells via semi-qRT-PCR. We essentially observed the same result as that of the qRT-PCR analysis; namely, we did not observe a clear increase in the expression of these Hh ligand genes in *shha*-deficient epicardial cells.

*5) The inability of detecting tagRFP is somewhat disappointing. I agree that low level of expression of the endogenous gene in epicardium may very well be a contributing factor. However, shha is highly expressed in the floor plate and notochord during development. It is unclear if the authors detected tagRFP signal in developing embryos after global inversion by Cre mRNA injection.*

We observed only weak TagRFP expression in floor plate cells in some embryos after the global inversion of the gene trap cassette. We considered this expression inconclusive, and we also examined other tissues to confirm the expression of TagRFP from the inverted cassette. We detected TagRFP expression most convincingly in the ganglion cell layer (GCL) of the retina, in which robust *shha* mRNA expression was reported previously (Shkumatava A, et al. Development 2004), and presented the result in Figure 2—figure supplement 3.

It is unclear why TagRFP expression does not fully recapitulate endogenous *shha* mRNA expression. We currently speculate that this is attributable to inefficient cleavage of the 2A peptide sequence. A glycine-serine-glycine spacer was recently revealed to significantly improve the cleavage efficiency of the 2A peptide (Wang Y, et al. Scientific Report2015), but we were unable to use this spacer because it was reported after the construction of *Zwitch*. We will address this issue in a future study to improve the visualization of mutant cells using *Zwitch*.

We revised the manuscript as follows:

“We detected TagRFP expression in cells in the ganglion cell layer (GCL) in the retina (arrowheads, Figure 2—figure supplement 3), in which robust *shha* expression was previously reported (Shkumatava et al., 2004). […] It is of interest to examine whether TagRFP expression would be enhanced in the target tissues by modifying the 2A sequence of *Zwitch* to include a glycine-serine-glycine spacer (GSG), which was recently found to significantly improve the cleavage efficiency of the 2A peptide (Wang et al., 2015b).”

*6) PCRs unfortunately do not include a positive control proving that samples that show no amplification contained DNA. For example, in 1F and 2B, C, E primers amplifying the wild-type shh locus or cDNA should be included.*

We appreciate the reviewer’s comment. We repeated the experiment described in Figure 1 with an appropriate control. Regarding the experiments described in Figure 2, and E, we had performed the experiments with the controls, but we did not include the data in the original manuscript. We have included the control data in each figure in the revised manuscript.

*7) Figure 3: it's unclear how the authors identified ct/ct embryos. Methods indicate that they did not raise homozygous adults, which then were incrossed. Rather it seems the experiments were done on individual embryos genotyped from an incross of ct/+ fish? Please clarify and make this also clear in the manuscript text. If the latter: an example of a genotyping PCR should be shown that can distinguish ct/+ from ct/ct embryos with and without Cre.*

In this experiment, we used *shha^ct/ct^*embryos prepared from incrosses of *shha^ct/+^*fish and genotyped the embryos individually via PCR. We presented examples of the PCR genotyping results of embryos from incrosses of *shha^ct/+^*fish and crosses of *shha^ct/+^*fish with *tcf21:CreER; shha^ct/+^*fish in Figure 3—figure supplement 1. Furthermore, we clarified that the screening was performed using individual embryos in the manuscript text as follows:

“To validate that *shha^ct^*was a loss-of-function allele, we obtained *shha^ct/ct^*embryos from incrosses of *shha^ct/+^*fish and individually genotyped the embryos using PCR (Figure 3—figure supplement 1).”

“Next, we crossed *shha^ct/+^*fish with *shha^ct/+^*fish carrying the epicardium-specific inducible Cre transgene *Tg(tcf21:CreER) (tcf21:CreER*) (Kikuchi et al., 2011) and obtained *tcf21:CreER; shha^ct/ct^*embryos after PCR genotyping of individual embryos (Figure 3—figure supplement 1).”

“To analyze epicardial cell development and *raldh2* expression in epi-KO hearts, we crossed *tcf21:CreER; shha^ct/+^*with *shha^ct/+^*fish carrying the epicardium-specific DsRed2 reporter transgene *Tg(tcf21;DsRed2) (tcf21:DsRed2*) (Kikuchi et al., 2011) and obtained *tcf21:DsRed2; tcf21:CreER; shha^ct/ct^*embryos after PCR genotyping of individual embryos.”

In the revised manuscript, we genotyped embryos from incrosses of *ubb:Cre-GFP; shha^ct/+^*fish to address comment #8. Genotyping was similarly performed on these embryos, and the result is presented in Figure 3—figure supplement 2.

Moreover, we have significantly revised the subsection “Genotyping” and included a new subsection “Sample preparation for embryo genotyping and analysis” in the Materials and methods. We have provided more detailed descriptions of the methods by which embryos were processed for genotyping PCR.

*8) To prove that the inverted ct allele is a loss-of-function allele of shh, the authors should establish a non-mosaic line of heterozygous fish containing the mutagenic allele and analyze the phenotype of embryos derived from incrosses of these and quantify shh RNA expression (protein quantification is probably not possible due to non-availability of anti-shha antibodies?)*

We thank the reviewer for this important suggestion. However, establishing a new non-mosaic line containing the heterozygous mutagenic allele and analyzing its offspring could not be performed within the given period for this revision. Instead, we performed the following experiment, which we believe addresses this issue. We presented the result as Figure 3—figure supplement 2 in the revised manuscript.

We established *shha^ct/+^*fish carrying the transgene *Tg(ubb:iCRE-GFP)*, in which the expression of codon-improved Cre (iCRE) DNA is expressed by the *ubiquitin B (ubb*) promoter (Figure 2—figure supplement 1). Although the *ubb:Cre-GFP; shha^ct/+^*line is not the non-mosaic line that the reviewer suggested we establish, we suspect that the gene trap allele is inverted in all cells, including germ cells, of this line due to the strong and ubiquitous activity of the *ubb*promoter. Consistent with global inversion in this strain, PCR genotyping of 48 single embryos prepared from incrosses of this line showed that all embryos contained the wild-type and/or the inverted mutagenic allele, and no embryos maintained the non-mutagenic allele, irrespective of the presence of the Cre transgene (Figure 3—figure supplement 2). Thus, mosaicism with the mutagenic allele was virtually undetectable in the offspring of the *ubb:Cre-GFP; shha^ct/+^*line.

We analyzed the phenotype of Cre transgene-negative embryos carrying wild-type *shha* alleles, heterozygous mutagenic alleles, or homozygous mutagenic alleles. Pectoral fin development was normal in all wild-type embryos (all five analyzed fish were normal) and most heterozygous mutants (six of seven fish were normal), but severely hampered in all homozygous mutants (all three analyzed fish were abnormal) (Figure 3—figure supplement 2). We also performed semi-qRT-PCR analysis of *shha* expression in these embryos (Figure 3—figure supplement 2). Densitometric quantification of the PCR result demonstrated that *shha* expression was reduced in a concentration-dependent manner with the mutagenic allele (Figure 3—figure supplement 2). Together with the observation from the Cre DNA injection experiment (Figure 3), we believe these results strongly support our conclusion that the inverted *shha^ct^*allele is a loss-of-function allele of *shha*. We have described this result in the revised manuscript as follows:

“To confirm this result, we established *shha^ct/+^*fish carrying the transgene *Tg(ubb:iCRE- GFP) (ubb:Cre-GFP*), in which codon-improved Cre (iCRE) DNA was expressed by a strong, ubiquitously expressed ubiquitin B (*ubb*) promoter (Mosimann et al., 2011) (Figure 2—figure supplement 1). […] We also performed semi-qRT-PCR analysis of *shha* expression and confirmed that its expression was reduced to nearly 50% of WT levels in the heterozygous mutants and to an undetectable level in the homozygous mutants (Figure 3—figure supplement 2).”

Moreover, we described the PCR genotyping of the mutagenic *shha* allele and Cre transgene in “Genotyping” and generation of the *ubb:Cre-GFP* line in “Zebrafish” in the Materials and methods.

*9) 4C: sample size and frequency of the edema phenotype needs to be given.*

We analyzed vehicle- or 4-HT–treated *tcf21:CreER; shha^ct/ct^*embryos as shown in Figure 4. We did not observe cardiac edema in the vehicle-treated embryos examined (0 abnormal in 8 analyzed) but found severe cardiac edema in six 4-HT–treated embryos (6 abnormal in 8 analyzed; p< 1.0 × 10^-8^, Fisher’s exact test). We have included this information in Figure 4 and the Figure 4 legend in the revised manuscript.

*10) Is the heart edema and reduced CM-number phenotype also observed in non-conditional shha mutants? If not (I could not find reports describing it), the authors should present a hypothesis why this would only be seen in the conditional mutant.*

We observed severe cardiac edema and reductions in cardiomyocyte cell numbers in the conditional mutant heart at 96 and 120 hpf (Figure 4). The cardiac edema phenotype was not evident or extremely weak at 72 hpf in the conditional mutant heart. We analyzed the global *shha* mutant embryos at 72 hpf (Figure 3) but not at later time pointes, as proper characterization was nor possible due to developmental abnormalities associated with global inactivation of *shha* expression. We suspect that inactivation of *shha* expression in the epicardium leads to heart defects mainly at later developmental stages, and such phenotypes may be obscured in non-conditional mutant embryos by pleiotropic effects of global *shha* inactivation. We included this interpretation in the revised manuscript as follows:

“Cardiac edema was unclear or extremely weak in epi-KO hearts at 72 hpf, suggesting that epicardial inactivation of *shha* expression leads to heart defects at later developmental stages. We could not determine whether a similar cardiac phenotype was also observed in the global *shha* mutant embryos, as proper characterization was not possible due to the pleiotropic effects of global inactivation of *shha* expression at later time points.”

*11) This article aims to provide a novel construct for using Cre-mediated recombination to conditionally ablate genes in zebrafish. Overall, the ability to efficiently introduce targeted mutations would be an important advance for functional work in zebrafish. The utility of this strategy seems contingent on the efficiency of homologous recombination. The manuscript would benefit by including an additional description of how the efficiency of their strategy compares with other published approaches. Is this really any better than what is out there?*

The targeting efficiency and germline transmission rate of genome editing was likely affected by a number of experimental parameters such as the targeted gene loci, inserted DNA size, and donor vector design. Therefore, it is difficult to directly compare the efficiency achieved in our study to those in other studies. Rather, we included one paragraph to discuss how other published approaches can be used with *Zwitch* to facilitate functional work in zebrafish as follows:

“After pioneering studies demonstrated the editing of zebrafish genome with long dsDNA (Bedell et al., 2012; Zu et al., 2013), several studies reported the insertion of large DNA fragments at defined zebrafish gene loci via DSB-mediated HR (Hisano et al., 2015; Hoshijima et al., 2016; Shin et al., 2014; Zu et al., 2013) or homology-independent repair mechanisms (Auer et al., 2014; Kimura et al., 2014; Li et al., 2015). […] Although we could not efficiently induce DSBs in the *shha* intronic sequence using the CRISPR/Cas9 system, this approach may be useful for targeting *Zwitch* into other genes.”

*12) For a methods paper, little information is given on the approach used for designing the left arm and right arm homology sequences and the TALENs. For a general audience, this needs to be expanded. Additionally, the Materials and methods should be expanded to include all details (including relevant product numbers) for vector construction and the specific steps that would be required for using the Zwitch construct to target other genes, such that anyone with reasonable molecular biology skills could follow step-by-step.*

We have included a subsection titled “Targeting *Zwitch* into other genes” in the Materials and methods, in which we described the design of the TALENs and homology arms and explained key steps for *Zwitch* construction for other genes, including details of the reagents used.

In addition, we mentioned this in the manuscript text as follows:

“We explained key steps for using *Zwitch* for other genes in the subsection titled “Targeting *Zwitch* into other genes” in the Materials and methods.”